# Visualizing VDAC1 in live cells using a tetracysteine tag

**Johannes Pilic[1], Furkan E. Oflaz[1], Benjamin Gottschalk[1], Yusuf C. Erdogan[1], Wolfgang F. Graier [1,2], Roland Malli [1,2]***

**1** Gottfried Schatz Research Center, Molecular Biology and Biochemistry, Medical University of Graz, Graz, Austria, **2** BioTechMed Graz, Graz, Austria

\* roland.malli@medunigraz.at

## Abstract

The voltage-dependent anion channel 1 (VDAC1) is a crucial gatekeeper in the outer mitochondrial membrane, controlling metabolic and energy homeostasis. The available methodological approaches fell short of accurate visualization of VDAC1 in living cells. To permit precise VDAC1 imaging, we utilized the tetracysteine (TC)-tag and visualized VDAC1 dynamics in living cells. TC-tagged VDAC1 had a cluster-like distribution on mitochondria. The labeling of TC-tagged VDAC1 was validated with immunofluorescence. The majority of VDAC1-TC-clusters were localized at endoplasmic reticulum (ER)-mitochondria contact sites. Notably, VDAC1 colocalized with BCL-2 Antagonist/Killer (BAK)-clusters upon apoptotic stimulation. Using this new tool, we were able to observe VDAC1-TC at mitochondrial fission sites. These findings highlight the suitability of the TC-tag for live-cell imaging of VDAC1, shedding light on the roles of VDAC1 in cellular processes.

## Introduction

Fluorescent proteins (FPs) have become an indispensable tool to tag and visualize proteins in living cells [1]. However, the large size of FPs (~25 kDa), can potentially interfere with the function and localization of the tagged protein [2]. Such complications have been reported by GFP-tagging of the voltage-dependent anion channel 1 (VDAC1), a crucial transporter of metabolites and ions in the outer mitochondrial membrane [3, 4]. Correct insertion of VDAC1 into the membrane depends on the location and length of the protein tag [5]. It has been shown that the presence of a C-terminal tag of eight or more residues can lead to mistargeting of VDAC1 [3–5], most likely by interfering with the recognition of the targeting signal by the mitochondrial import receptor Tom20 [6]. Based on this knowledge, we hypothesized that a fusion tag shorter than eight residues would allow live-cell imaging of correctly localized VDAC1.

The tetracysteine (TC)-tag, consisting of only six residues (CCPGCC), has emerged as a promising tool to visualize protein dynamics in living cells [7]. The TC-tag becomes fluorescent upon reacting with fluorogenic dyes [8], such as the **fl**uorescein **ars**enical **h**airpin binder-**e**thane**di**thiol (FlAsH-EDT$_2$) or the red-shifted analog **re**sorufin **ars**enical **h**airpin binder-

**Funding:** The research was supported by the Molecular Medicine PhD program of the Medical University of Graz and the FWF (Austrian Science Fund: DK-MCD W1226 to W.F.G and I3716-B27 to R.M.). The funders had no role in study design, data collection and analysis, decision to publish, or preparation of the manuscript.

**Competing interests:** The authors have declared that no competing interests exist.

**e**thane**di**thiol (ReAsH-EDT$_2$). Here, we aim to assess the functionality and suitability of the TC-tag as a tool to image VDAC1 in live cells. We anticipate that incorporating the TC-tag will enable the precise visualization of VDAC1 localization, trafficking, and colocalization with organelles.

## Materials and methods

### Cell culture

HeLa S3 cells, a clonal derivative of the parent HeLa cell line, were obtained from ATCC and authenticated with STR analysis. HeLa cells were cultured in Dulbecco's modified Eagle's medium (DMEM D5523, Sigma-Aldrich) supplemented with 10% FCS, 10 mM NaHCO$_3$, 50 U/mL penicillin-streptomycin, 1.25 μg/mL amphotericin B and 25 mM HEPES; pH was adjusted to 7.45 with NaOH. Cells were grown in a humified atmosphere of 5% CO$_2$ at 37˚C.

### Transfection

Cells were seeded in 6-well plates on 30 mm glass coverslips (Paul Marienfeld GmbH & Co. KG, Lauda-Königshofen, Germany) and transfected using PolyJet (SignaGen Laboratories). Per well, 3 μl of PolyJet reagent was mixed with 1 μg of plasmid DNA in 100 μl of DMEM devoid of serum and antibiotics. The details of plasmids are provided in Table 1. The transfection mixture was added to 1 ml of culture medium for 8 hours and was then replaced with 2 ml of culture medium. Imaging was performed 24–48 h after transfection.

### Imaging of subcellular protein dynamics

Before imaging, cells were put into a storage buffer, which was composed of 135 mM NaCl, 5 mM KCl, 2 mM CaCl$_2$, 1 mM MgCl$_2$, 10 mM HEPES, 2.6 mM NaHCO$_3$, 0.44 mM KH$_2$PO$_4$, 0.34 mM Na$_2$HPO$_4$, 10 mM D-glucose, 2 mM L-glutamine, 1X MEM amino, 1X MEM vitamins, 1% penicillin-streptomycin and 1% Amphotericin B; pH was adjusted to 7.45 with NaOH.

High-resolution imaging was performed with an array confocal laser scanning microscope (Axiovert 200 M, Zeiss) equipped with a 100×/1.45 NA oil immersion objective (Plan-Fluor, Zeiss) and a Nipkow-based confocal scanner unit (CSU-X1, Yokogawa Electric Corporation). Laser light of diode lasers (Visitron Systems, Pucheim, Germany) served as the excitation light source: CFP, GFP, and RFP fusion constructs were excited with 445, 488, and 561 nm lasers, respectively. Emission light was captured with a CoolSNAP HQ2 CCD Camera (Photometrics

**Table 1. List of plasmids used in this study.**

| Name | Description | Reference |
|---|---|---|
| GFP-VDAC1 | GFP fused to the N-terminus of VDAC1 | Addgene #211735 |
| VDAC1-GFP | GFP fused to the C-terminus of VDAC1 | Addgene #211734 |
| VDAC1-TC | TC-tag fused to the C-terminus of VDAC1 | Addgene #211733 |
| mitoDsRed | Mitochondrial matrix marker | Addgene #44386 |
| mito-sfGFP | Mitochondrial matrix marker | [9] |
| Hexokinase1-GFP | Hexokinase1 marker | Addgene #21917 |
| mCh-ER3 | Luminal ER marker | Addgene #55041 |
| mCh-Sec61β | Membrane ER marker | Addgene #49155 |
| mtHyPer7 | Mitochondrial hydrogen peroxide indicator | Addgene #136470 |
| mCh-Mff | Mff marker | Addgene #157760 |
| GFP-BAK | BAK marker | Addgene #32564 |

Tucson, Arizona, USA) using the emission filters ET460/50m, ET525/36m, and ET630/75m (Chroma Technology Corporation) for CFP, GFP, and RFP fusion constructs, respectively. FlAsH was excited like GFP for two-color imaging and like YFP for three-color imaging (excitation laser: 514 nm; emission filter ET535/30m). ReAsH was excited like RFP.

Super-resolution imaging was performed with a structured illumination microscope (Nikon) equipped with a 100×/1.49 NA oil immersion objective (CFI Aopchromat TIRF, Nikon), standard filter sets, and two iXon EMCCD cameras (Andor). FlAsH labeled VDAC1-TC and mCh-Sec61β were excited with 488 and 561 nm lasers, respectively.

## Statistics and image analysis

Samples represent technical replicates. The sample size was chosen to adequately address the research objective while ensuring sufficient statistical power. Image analysis was performed with Fiji software. Z-stack images with a step size of 200 nm were deconvoluted and background-subtracted using a rolling ball radius of 50 to 300 pixels. For colocalization analysis, the TrackMate plugin was used to identify and quantify colocalization between VDAC1-clusters and the ER. For co-localization analyses, the ImageJ plugin coloc2 was used to measure the Pearson coefficient and Manders' coefficient. To assess ER morphology, 3D images of a luminal ER marker were thresholded using Sauvola local threshold and Li global threshold. ER morphology parameters were assessed with the ImageJ plugin 3D Manger. To assess mitochondrial morphology, 2D images of a mitochondrial matrix marker were thresholded using the Otsu method. For the Aspect Ratio, the ratio of the longest and shortest axes of a mitochondrial matrix marker was calculated. For the Form Factor, the following formula was used:

$$Form\ Factor = \frac{Perimeter^2}{4\pi \times \text{Area}}$$

## VDAC1-TC labeling

Cells expressing tetracysteine-tagged VDAC1 (VDAC1-TC) were labeled with FlAsH-EDT$_2$ or ReAsH-EDT$_2$ (Cayman chemical, Michigan USA) in storage buffer for 15 min at 37˚C. Dye concentrations are indicated in figure legends. The cells were washed with 100 μM BAL (2,3-dimercaptopropanol or British anti-Lewisite) in storage buffer for 10 min at 37˚C and kept in storage buffer before confocal imaging.

## Membrane potential measurements

Cells were stained with 25 nM TMRM (Molecular Probes, Invitrogen) for 15 min at 37˚C before membrane potential measurements at Nikon eclipse Ti2 microscope. The microscope was equipped with a 40×/1.15 NA water immersion objective (CFI Apochromat, Nikon), standard filter sets, and two Kinetix Scientific CMOS cameras (Photometrics). TMRM was excited with 580 nm light from pE-800 (CoolLED). Per repeat, 6x6 images were stitched, background subtracted, and thresholded with the Triangle method before measuring TMRM intensity. For dynamic mitochondrial membrane potential measurements, the ratio between mitochondrial to nuclear TMRM intensity was calculated and the minimum ratio induced by FCCP was normalized to 1.

## Perfusion of cells during live cell imaging

Transfected cells on 30 mm coverslips were put in a PC30 perfusion chamber (NGFI, Graz, Austria) and perfused at a flow rate of approximately 1 ml per minute with a gravity-based perfusion system (PS9, NGFI). Glucose buffer was composed of 135 mM NaCl, 5 mM KCl, 2 mM

CaCl₂, 1 mM MgCl₂, 10 mM HEPES, and 10 mM D-glucose; pH was adjusted to 7.45 with NaOH. The glucose-free buffer contained 10 mM D-mannitol instead of glucose.

### Immunofluorescence

HeLa cells transfected with VDAC1-TC were seeded in 6-well plates on 30 mm coverslips to reach a confluency of ~80%. Cells were then taken from the incubator and put in storage buffer containing 1 μM FlAsH for 15 min at 37°C. Cells were washed with 100 μM BAL for 10 min. Then cells were washed twice with PBS and fixed with 4% paraformaldehyde at room temperature for 15 min. Then cells were washed 3x with PBS and blocked with 5% BSA in PBS for one hour while mildly shaking. Then the blocking buffer was replaced with 1:1000 diluted VDAC1 mouse monoclonal antibody (abcam, #15895) in PBS with 5% BSA. Cells were incubated with the primary antibody overnight at 4°C, mildly shaking. On the next day, primary antibody solution was removed, cells were washed 3x with PBS, and secondary antibody solution containing Alexa Fluor 568 goat anti-mouse (Thermo Fisher Scientific) at a dilution of 1:2.500 was added. Cells were incubated for 2 hours in the dark at room temperature, mildly shaking. Finally, cells were washed 3x for 5 minutes with PBS, Vectashield antifade mounting medium (Vectorlabs), and the coverslips were sealed with nail polish.

### Seahorse measurements

Sells were seeded on XF96 polystyrene cell culture microplates (Agilent, Seahorse) to a confluency of 100%. Before the measurement cells were washed and incubated in XF assay medium supplemented with 1 mM sodium pyruvate, 2 mM glutamine, and 5.5 mM D-glucose. The ratio of oxygen consumption rate (OCR, pmol O2/min) to extracellular acidification rate (ECAR, mpH/min) was measured with XF96 extracellular flux analyzer (Agilent, Seahorse).

### Chemicals

To stain mitochondria, cells were incubated with 166 nM MitoTracker Red CMXRos (Thermo Fisher Scientific) in storage buffer for 15 min at 37°C. To induce apoptosis, cells were incubated with 10 μM staurosporine (Sigma) for one hour in storage buffer. To induce mitochondrial fission, cells were perfused with 2 μM FCCP (Sigma) or 4 μM Ionomycin (Abcam) for 10 min in the glucose buffer.

### Generation of 3D structures of VDAC1-fusion constructs

3D structures of GFP-VDAC1, VDAC1-GFP, and VDAC1-TC were predicted using Colab-Fold v1.5.2-patch, a modified version of AlphaFold2 that integrates MMseqs2 for sequence alignment and structure prediction. 3D structures were visualized using UCSF Chimera.

## Results

### Mistargeting of VDAC1 is induced by N- and C-terminal fusion of GFP

To assess whether VDAC1 can fold properly when fused to FPs with a flexible linker, we used AlphaFold2 to generate 3D structures for GFP-VDAC1 and VDAC1-GFP (Fig 1A). The 3D structures showed favorable folding patterns for both constructs, suggesting that the fusion of FPs does not compromise the quarternary structure of VDAC1.

To validate the correct targeting of these constructs, we imaged HeLa cells transfected with GFP-VDAC1 or VDAC1-GFP and stained with mitotracker red, a dye targeted to the inner mitochondrial membrane. However, we observed that in the majority of transfected cells, both constructs were predominantly aggregated in the cytosol (Fig 1B). We also observed cytosolic

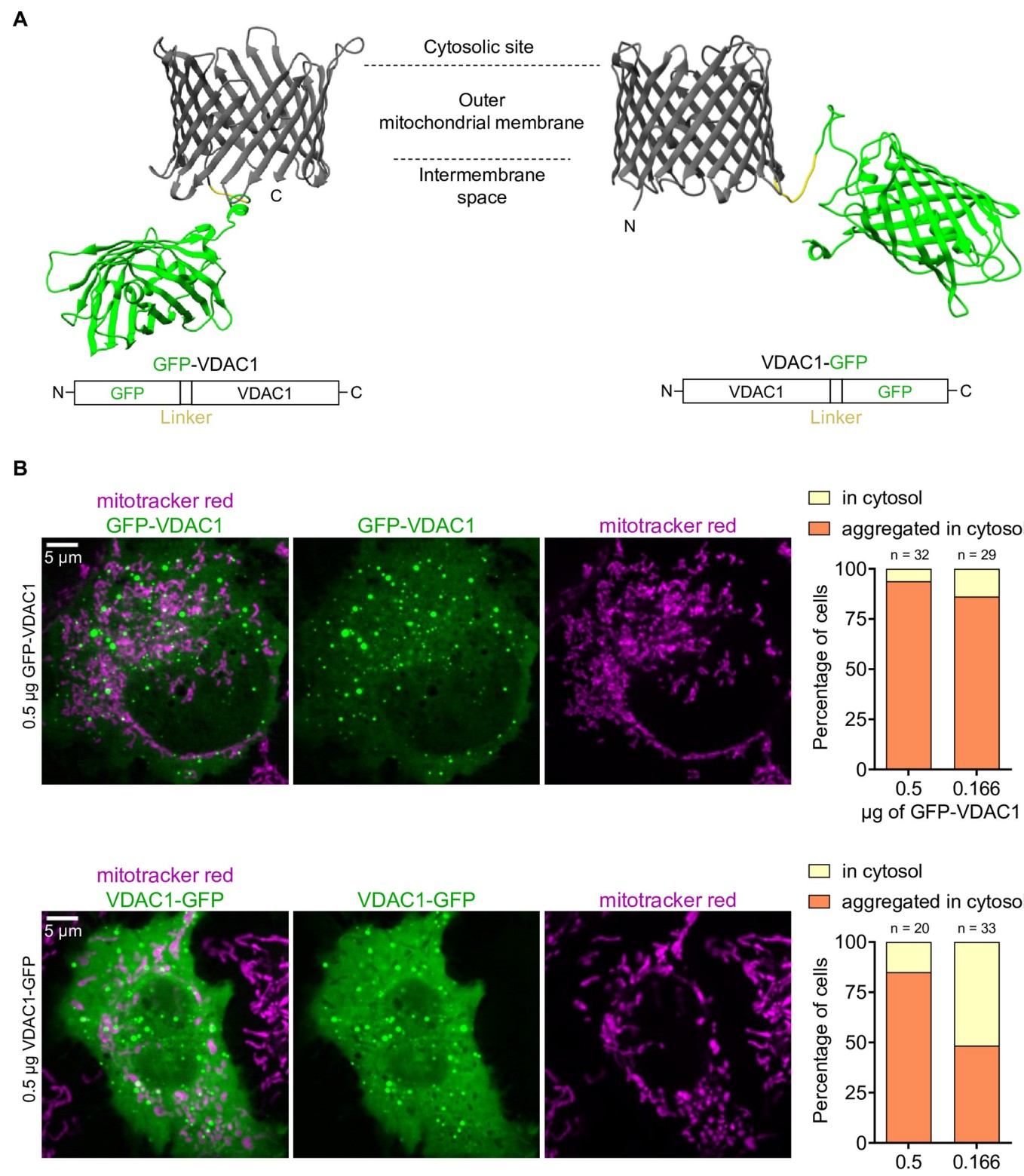

**Fig 1. Mistargeting of VDAC1 is induced by N- and C-terminal fusion of GFP.** (**A**) AlphaFold2-generated 3D structures and schematic representation of GFP-VDAC1 and VDAC1-GFP fusion constructs. (**B**) Confocal images of HeLa cells stained with mitotracker red and transfected with GFP-VDAC1 (top panel) or VDAC1-GFP (bottom panel). Bar graphs (right) show the percentage of HeLa cells with different expression phenotypes of GFP-VDAC1 (top) or VDAC1-GFP (bottom) using different amounts of plasmid DNA.

localization of the constructs without apparent aggregates (S1A Fig). Using less plasmid DNA to reduce potential overexpression artifacts, we observed a reduced percentage of cells with aggregated FP-tagged VDAC1 (Fig 1B), but we did not observe mitochondrial localization (S1B Fig). These data indicate that the fusion of GFP to VDAC1 leads to mistargeting.

## Short tetracysteine-tag allows to visualize VDAC1-clusters on mitochondria

Since VDAC1 tends to mistarget when attached to a C-terminal tag of eight or more residues [5], we opted to fuse VDAC1 with a six-residue long tetracysteine (TC)-tag. We choose to tag the C-terminus of VDAC1, as the N-terminal domain is known to interact with numerous proteins [10]. To visualize VDAC1-TC, we used two chemical dyes: FlAsH-EDT$_2$ (Fig 2A) and the red-shifted ReAsH-EDT$_2$ (Fig 2C). As we imaged HeLa cells coexpressing FlAsH labeled VDAC1-TC and mitoDsRed, an FP targeted to the mitochondrial matrix, we observed two distinct labeling patterns: Transfected cells showed a cluster-like distribution of VDAC1-TC around mitochondria (Fig 2A). Untransfected cells showed homogenous labeling of mitochondria by FlAsH, indicating unspecific mitochondrial accumulation of the dye (Fig 2A). Importantly, transfection with mitoDsRed alone did not lead to a cluster-like distribution of FlAsH (S2A Fig), suggesting that the dye alone does not cluster on mitochondria. Due to structural similarities of FlAsH with Rhodamine123 (S2B Fig), a common potentiometric mitochondrial dye, we assumed that unspecific binding of FlAsH to mitochondria could be reduced with carbonyl cyanide-p-trifluoromethoxy-phenylhydrazone (FCCP), a mitochondrial uncoupler. Indeed, treating cells with FCCP during labeling significantly reduced the binding of FlAsH to mitochondria compared to FCCP untreated cells (S2B Fig), suggesting unspecific binding of FlAsH is caused by mitochondrial membrane potential. The classical washing buffer for the TC-labeling approach, British anti-Lewisite (BAL), also reduced the binding of FlAsH to mitochondria, but not as much as FCCP (S2B Fig). Another possibility is that FlAsH and ReAsH, due to their aromatic structure, may non-specifically bind cysteine-rich domains in mitochondrial proteins [11]. To estimate the potential side effects of FlAsH on mitochondrial energy status, we assessed mitochondrial membrane potential with tetramethylrhodamine methyl ester (TMRM) and the ratio between oxygen consumption rate (OCR) to extracellular acidification rate (ECAR). We did not observe significant changes in TMRM intensity (S2C Fig), indicating unaffected mitochondrial membrane potential and OCR/ECAR ratio (S2D Fig) upon labeling with FlAsH or washing with BAL.

FlAsH consist of aromatic rings with delocalized electrons that can generate reactive oxygen species (ROS) when irradiated, potentially affecting mitochondrial membrane potential. Therefore, we tested whether FlAsH stain affects mitochondrial membrane potential during imaging. FlAsH staining did not affect mitochondrial membrane potential over time, compared to unstained cells (S2E Fig). VDAC1-TC transfected cells had a lower basal mitochondrial membrane potential compared to untransfected cells (S2F Fig), however, irradiation did not lead to ROS generation in both groups (S2G Fig). These data indicate that FlAsH staining does not affect ROS production and mitochondrial membrane potential during time-lapse imaging.

To verify that the FlAsH signal in cells transfected with VDAC1-TC is in fact VDAC1 we performed immunofluorescence of VDAC1. We observed a high colocalization between VDAC1 antibody labeling and VDAC1-TC FlAsH signal (Fig 2B), confirming that the FlAsH signal corresponds to VDAC1-TC. In untransfected cells, we could not distinguish between endogenous VDAC1 labeling and secondary antibody control (S2J Fig), suggesting that VDAC1 antibody is not sensitive enough to detect endogenous VDAC1.

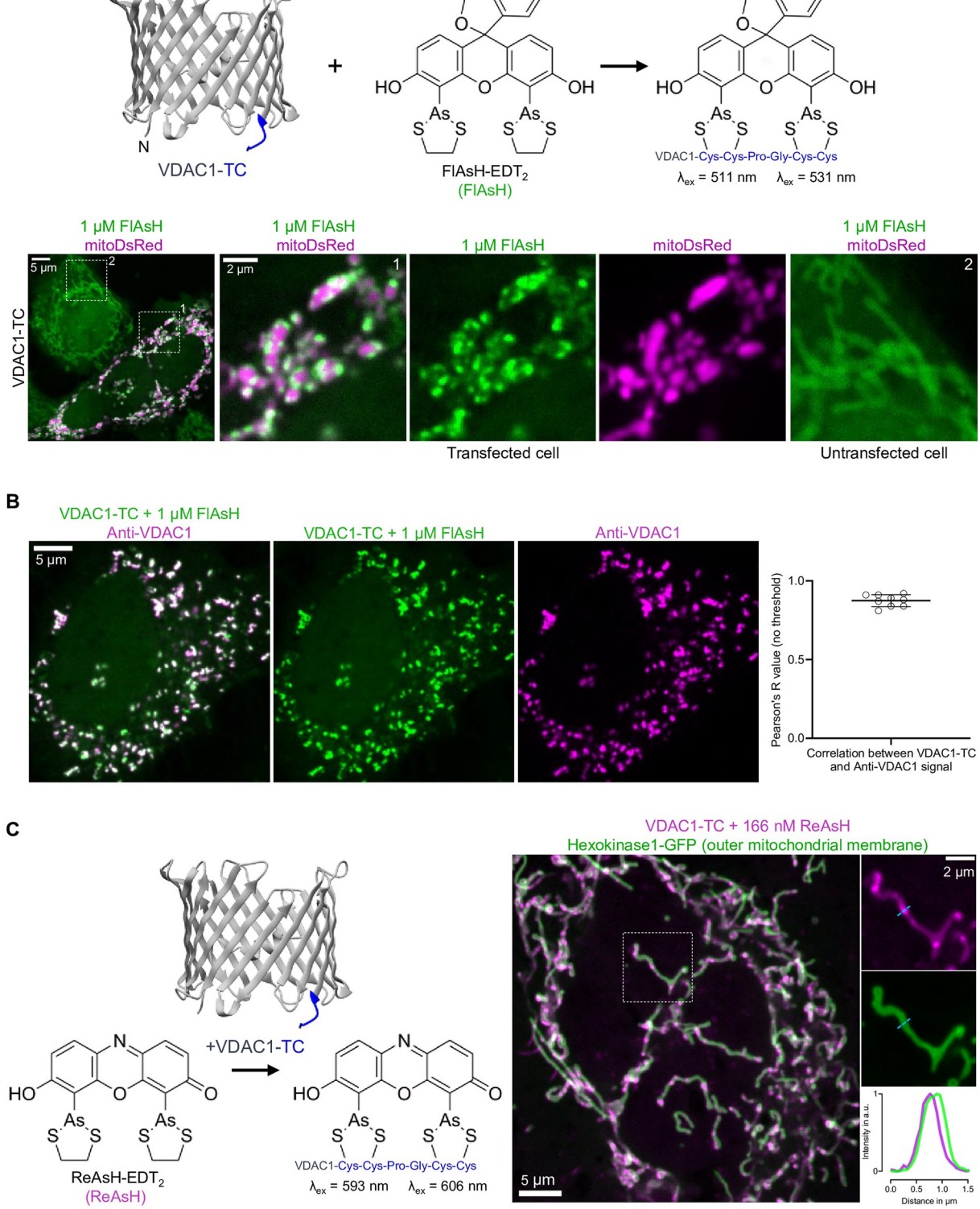

**Fig 2. Short tetracysteine-tag allows to visualize VDAC1-clusters on mitochondria.** (**A**) 3D structure of VDAC1-TC and chemical structure of FlAsH (top). Confocal images of HeLa cells transfected with VDAC1-TC and mitoDsRed; cells were labeled with 1 μM FlAsH for 15 min and washed with 100 μM BAL for 10 min (bottom). Left image shows an overview of transfected and untransfected cells, and the numbered dashed squares are magnified on the right. (**B**) Immunofluorescence images of a HeLa cell expressing VDAC1-TC; cell was labeled with 1 μM FlAsH for 15 min (green) and anti-VDAC1 (magenta). Graph on the right shows the Pearson correlation between VDAC1-TC and Anti-

VDAC1 signal. (**C**) 3D structure of VDAC1-TC and chemical structure of ReAsH (left). Confocal images of a HeLa cell expressing VDAC1-TC and Hexokinase1-GFP; cells were labeled with 166 nM ReAsH for 15 min and washed with 100 μM BAL for 10 min (right). Dashed lines represent the line scan graph on the right and show the relative fluorescence intensity of Hexokinase1-GFP (green) and VDAC1-TC (magenta) along the length of the line.

Next, we tested the efficacy of our tool over time. As a proof of principle, we visualized ReAsH-labeled VDAC1-TC with high temporal resolution (S1 Video) and FlAsH-labeled VDAC1-TC over hours (S2 Video). VDAC1-TC-clusters remained in size, form, and localization over time (S1 and S2 Videos), suggesting low motility of VDAC1 in the outer mitochondrial membrane. Additionally, we tested whether overexpression of Hexokinase 1 (HK1), which has been shown to interact with VDAC1 on the outer mitochondrial membrane (OMM) [12], affects the cluster-like distribution of VDAC1-TC. Using HK1-GFP and ReAsH labeled VDAC1-TC we observed that VDAC1-clusters localized on the OMM (Fig 2C), suggesting that HK1 overexpression does not affect the subcellular localization of VDAC1-TC-clusters.

## VDAC1-TC-clusters are localized at ER-mitochondria contact sites

Since VDAC1 has been described to localize to ER-mitochondrial contact sites [13–15], we examined whether VDAC1-TC-clusters colocalize with the ER in living cells. For this purpose, we imaged HeLa cells coexpressing VDAC1-TC with the luminal ER-marker mCh-ER3 and mitoCFP. The majority of VDAC1-TC-clusters indeed localize to ER-mitochondrial contact sites (Fig 3A), with 63.6% of VDAC1-TC-clusters colocalizing with the ER (Fig 3B). To further validate that VDAC1-TC is localized at ER-mitochondrial contact sites, we performed colocalization analyses. The Pearson correlation between VDAC1-TC and mCh-ER was significantly reduced when the VDAC1-TC images were rotated by 90 degrees, with consistent changes in Manders' tM1 and tM2 (S3A Fig). Similarly, we found analogous outcomes for VDAC1-TC and mitoCFP correlations (S3B Fig). To further substantiate the localization of VDAC1-TC within ER-mitochondria contacts, we used super-resolution microscopy and mCh-Sec61β, an ER membrane marker. We observed clear colocalization of VDAC1-TC and mCh-Sec61β (Fig 3C), confirming that VDAC1-TC is localized at contact sites between the ER and mitochondria.

## VDAC1-TC-clusters colocalizes with BAK-clusters that form in response to stress

VDAC1 has been suggested to interact with BCL-2 Antagonist/Killer (BAK) [16]. To explore this dynamic interaction, we imaged HeLa cells coexpressing VDAC1-TC with GFP-BAK. BAK is a proapoptotic protein residing as an inactive monomer on the OMM until triggered by cellular stress to cluster into pores. We found that glucose depletion rapidly triggered BAK-clustering (Fig 4). Quantifying the colocalization between BAK and VDAC1, we found that 39.1% of VDAC1-TC-clusters colocalized with BAK-clusters and 23.0% of BAK-clusters colocalized with VDAC1-TC-clusters during glucose depletion (Fig 4). BAK-clusters with similar colocalization to VDAC1 were observed upon treatment with a classical pro-apoptosis activator, staurosporine (S4 Fig). These data indicate that a subset of VDAC1-TC-clusters colocalize with BAK-clusters during cellular stress in living cells.

## VDAC1-TC-clusters are observed at mitochondrial fission sites

We noted an increased fragmentation of mitochondria in cells expressing VDAC1-TC compared to untransfected cells. Therefore, we compared the mitochondrial morphology between

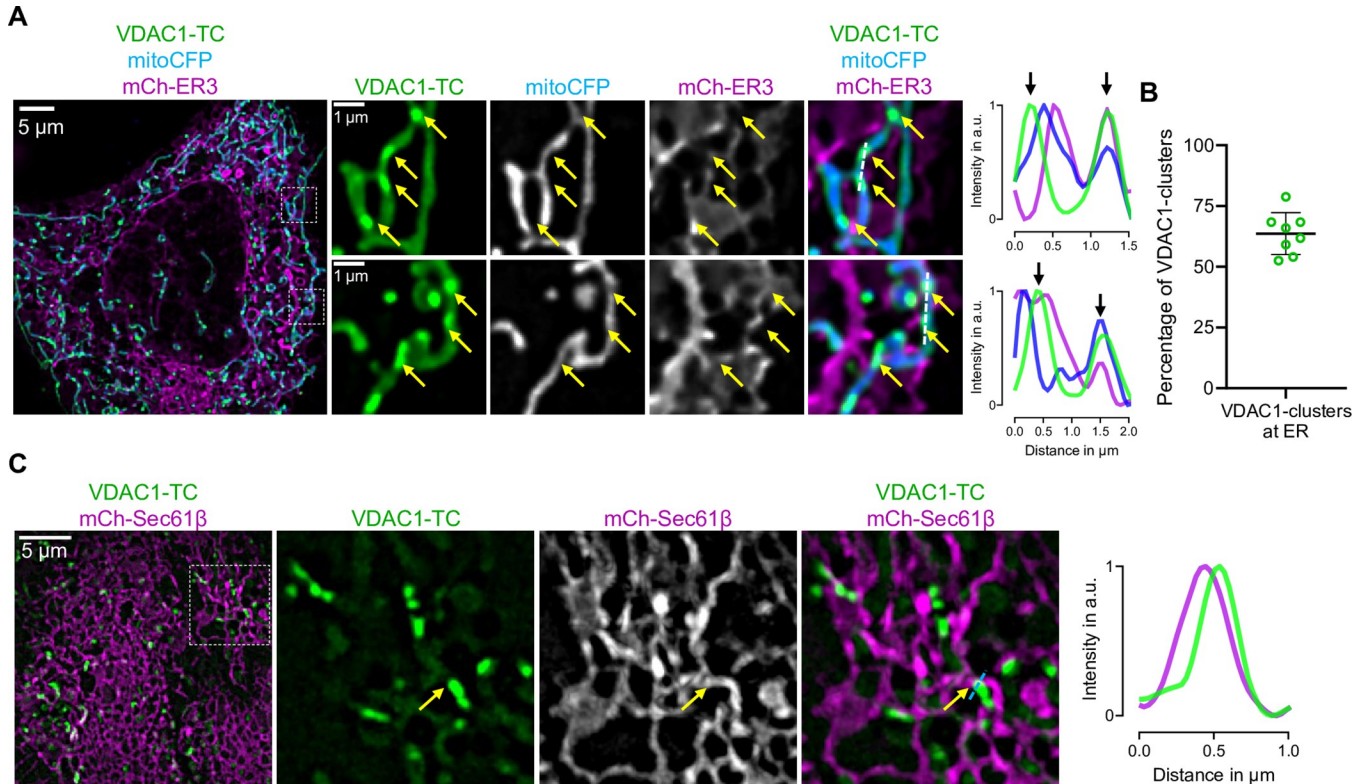

**Fig 3. VDAC1-TC-clusters are localized at ER-mitochondria contact sites.** (**A**) Confocal images of a HeLa cell expressing VDAC1-TC, mitoCFP, and mCh-ER3; the cell was labeled with 1 μM FlAsH for 15 min and washed with 100 μM Bal for 10 min. The left image shows an overview of the cell, and the dashed squares are magnified on the right side. Yellow arrows point to positions of VDAC1-clusters at ER-mitochondria contact sites. Dashed lines represent the line scan graphs on the right and show the relative fluorescence intensity of VDAC1-TC (green), mitoCFP (blue), and mCh-ER3 (magenta) along the length of the line. (**B**) The graph shows the percentage of VDAC1-clusters that colocalize with the ER (n = 8). (**C**) Structured illumination microscopy images of a HeLa cell expressing VDAC1-TC with mCh-Sec61β; cell was labeled with 1 μM FlAsH for 15 min and washed with 100 μM BAL for 10 min. Dashed line represent the line scan graph on the right and show the relative fluorescence intensity of VDAC1-TC (green) and mCh-Sec61β (magenta) along the length of the line. Arrow point to the position of a VDAC1-cluster at ER contact sites.

HeLa cells expressing mitoDsRed and HeLa cells coexpressing VDAC1-TC with mitoDsRed (Fig 5A). To assess mitochondrial morphology, we used Aspect Ratio (AR) and Form Factor (FF), which are indicators for mitochondrial swelling and branching, respectively. As evidenced by the significant reduction of AR and FF, VDAC1-TC overexpression was linked to mitochondrial fragmentation in HeLa cells (Fig 5A). Importantly, FlAsH or FlAsH in combination with BAL did not significantly affect mitochondrial morphology (S5A Fig). Since the mitochondria of most VDAC1-TC overexpressing cells were hyperfragmented, we assessed cell proliferation, as a marker of cellular health. We observed that cells overexpressing VDAC1-TC exhibited reduced cellular proliferation compared to sham-transfected cells (S5B Fig). The transfection efficiency of our construct was approximately 50% (S5C Fig). These data indicate that the hyperfragmentation induced by VDAC1-TC overexpression reduces cellular fitness.

Next, we visualized VDAC1-TC during mitochondrial fission events. We observed that VDAC1-TC-clusters were localized to sites of mitochondrial fission when cells were treated with FCCP (Fig 5B), a potent inducer of mitochondrial fission [17]. We assessed the presence of VDAC1-TC clusters at fission sites with the mitochondrial fission receptor mitochondrial fission factor (Mff), using mCh-Mff as a marker. We observed that VDAC1-TC-clusters were localized to Mff-clusters when cells were treated with FCCP (S5D Fig). Finally, we induced

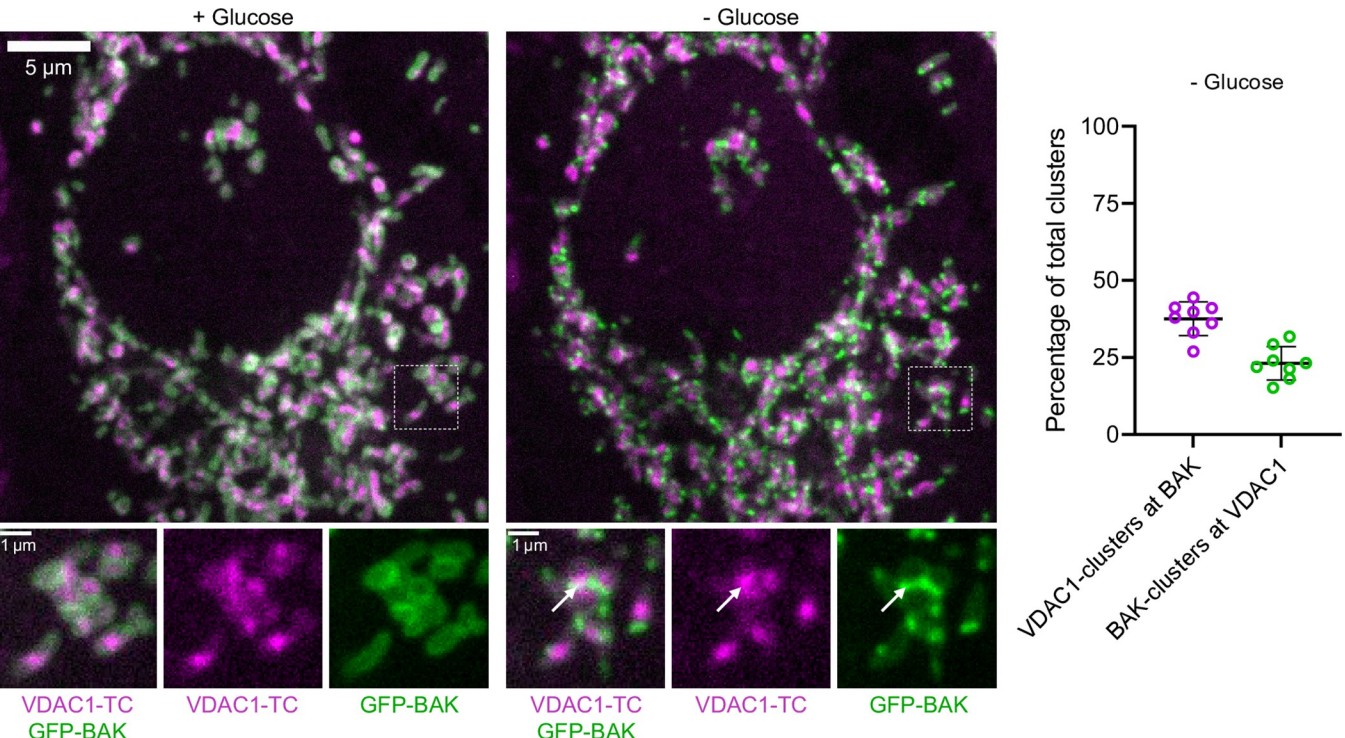

**Fig 4. VDAC1 colocalizes with BAK-clusters that form in response to stress.** Confocal images of a HeLa cell expressing VDAC1-TC and GFP-BAK with 10 mM glucose (left) and after 20 min of glucose depletion (middle); the cell was labeled with 333 nM ReAsH for 15 min. The first row shows an overview of the cell, and the dashed squares are magnified below. White arrows point to the contact site between VDAC1- and BAK-clusters. The graph shows the percentage of VDAC1-clusters that colocalized with BAK-clusters and the percentage of BAK-clusters that colocalized with VDAC1-clusters after 20 min of glucose depletion (right, n = 8 cells).

mitochondrial fission by increasing cellular $Ca^{2+}$ using ionomycin, a potent $Ca^{2+}$ ionophore. Ionomycin treatment caused strong mitochondrial fragmentation, with VDAC1-TC-clusters localizing to mitochondrial fission sites (Fig 5C). These data indicate that VDAC1 might be involved in mitochondrial fission.

## Discussion

Here, we demonstrate the efficacy of the short TC-tag in visualizing VDAC1 in living cells. By overcoming the limitations associated with larger FPs, the TC-tag offers a valuable tool for studying the roles of VDAC1 in cellular processes, such as ER-mitochondrial communication and mitochondrial dynamics.

Our data suggest that N- and C-terminal fusion of FPs leads to mistargeting and aggregation of VDAC1 in HeLa cells. This is in line with previous reports showing poor targeting of C-terminal FP-tagged rat and human VDAC1 in HeLa cells [3, 4]. In HEK-293T cells, there are studies showing rat VDAC1-GFP to be targeted to mitochondria [18, 19]. This mitochondrial localization of rat VDAC1-GFP could be due to different cell-type, transfection agent, or vector used in these studies. Using a short TC-tag, we could visualize VDAC1-clusters on mitochondria. This cluster-like distribution of VDAC1-TC has also been observed with FLAG-tagged VDAC1 and endogenously immunolabeled VDAC1 [20, 21]. We demonstrated that VDAC1-TC-clusters are predominately localized at ER-mitochondria contact sites using high- and super-resolution fluorescence microscopy. Since studies suggest that VDAC1 is

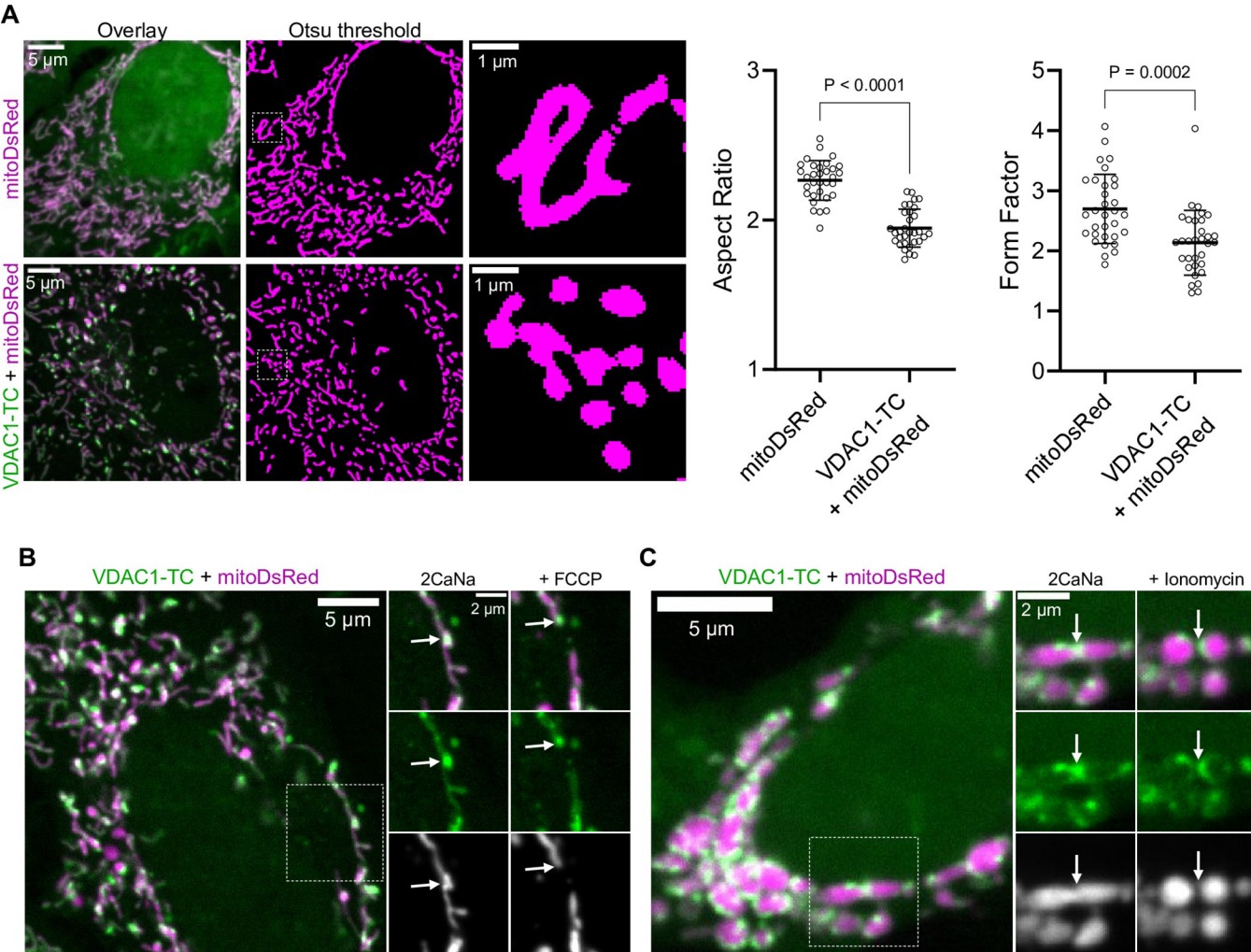

**Fig 5. VDAC1-TC-clusters are observed at mitochondrial fission sites.** (**A**) Confocal images of HeLa cells expressing mitoDsRed (top row) or VDAC1-TC with mitoDsRed (bottom row); cells were labeled with 1 µM FlAsH for 15 min and washed with 100 µM BAL for 10 min. The left column shows an overview of the cell, the middle column shows the Otsu thresholded mitoDsRed signal, and the dashed squares are magnified in the right column. Graphs on the right show the Aspect Ratio and Form Factor of thresholded images (n = 32 per group). The difference between groups was evaluated using unpaired t-test. Data are presented as mean ± SD. (**B**) Confocal images of a HeLa cells expressing VDAC1-TC and mitoDsRed; the cell was labeled with 1 µM FlAsH for 15 min and washed with 100 µM BAL for 10 min. On the left panel, an overview of the cell is shown, and the dashed square is magnified on the right side before and 10 min after perfusion with 2 µM FCCP. White arrows point to a VDAC1-cluster before and after mitochondrial fission. (**C**) Confocal images of a HeLa cell expressing VDAC1-TC and mitoDsRed; the cell was labeled with 1 µM FlAsH for 15 min and washed with 100 µM BAL for 10 min. On the right panel, an overview of the cell is shown, and the dashed square is magnified on the right side before and 10 min after perfusion with 4 µM Ionomycin. White arrows point to a VDAC1-cluster before and after mitochondrial fission.

localized at ER-mitochondrial contact sites [13–15], our findings indicate that TC-tagged VDAC1 reflects the same localization patterns as endogenous VDAC1.

A noteworthy discovery emerged as we noticed frequent colocalization between VDAC1- and BAK-clusters during glucose depletion or STS-induced apoptosis. While VDAC1 has been suggested to be involved in BAK-mediated apoptosis [16], concrete evidence of their colocalization was lacking until now.

We observed that VDAC1-TC-clusters localized to mitochondrial fission sites when cells were treated with FCCP or ionomycin. This spatial correlation suggests that VDAC1 might be involved in mitochondrial fission events. Interestingly, VDAC1 has been suggested to affect

mitochondrial dynamics. For example, knockdown of VDAC1 has been shown to prevent mitochondrial fragmentation induced by glutamate excitotoxity in cultured neurons [22].

In summary, we demonstrate the efficacy of the short TC-tag in visualizing the dynamics of VDAC1 in living cells. By overcoming the limitations associated with larger FPs, the TC-tag offers a valuable tool for studying the roles of VDAC1 in cellular processes, such as ER-mitochondrial communication and mitochondrial dynamics. Although currently, VDAC1-TC requires overexpression, which may have unintended consequences, the use of VDAC1-TC knock-in cell lines represents a natural progression of this work. The TC-tag may affect the conductivity, selectivity, and voltage-dependence of VDAC1, thus influencing its regulatory functions. Since VDAC1-TC is the only tool to image this essential protein in living cells, we believe this approach will provide insights into the role of VDAC1 in cellular processes.

## Supporting information

**S1 Fig. Mistargeting of VDAC1 is induced by N- and C-terminal fusion of GFP.** This figure shows examples of HeLa cells expressing FP-tagged VDAC1 without cytosolic aggregation. (PDF)

**S2 Fig. Short tetracysteine-tag allows to visualize VDAC1-clusters on mitochondria.** This figure shows confocal images of HeLa cells stained with FlAsH and analyses of mitochondrial function. (PDF)

**S3 Fig. VDAC1-TC-clusters are localized at ER-mitochondria contact sites.** This figure shows Pearson correlation and Manders' colocalization analysis between the ER, mitochondria and VDAC1-TC. (PDF)

**S4 Fig. VDAC1 colocalizes with BAK-clusters that form in response to stress.** This figure shows confocal images of HeLa cells treated with STS and colocalization analysis between VDAC1-TC and GFP-BAK clusters. (PDF)

**S5 Fig. VDAC1-TC-clusters are observed at mitochondrial fission sites.** This figure shows analyses of mitochondrial morphology and VDAC1-cluster localization in relation to Mff. (PDF)

**S1 Video. VDAC1-TC-clusters can be imaged with high temporal resolution.** This video shows a time-lapse video of a HeLa cell expressing VDAC1-TC and mito-sfGFP at 2-minute intervals. (AVI)

**S2 Video. VDAC1-TC-clusters can be imaged over hours.** This video shows a time-lapse video of a HeLa cell expressing VDAC1-TC and mito-sfGFP at 20-minute intervals. (AVI)

**S1 Data.** (DOCX)

## Acknowledgments

We thank Dr René Rost, Anna Schreilechner, and Mercedes Maier for cell seeding and technical assistance.

## Author Contributions

**Conceptualization:** Johannes Pilic, Wolfgang F. Graier, Roland Malli.

**Data curation:** Johannes Pilic, Furkan E. Oflaz, Benjamin Gottschalk, Roland Malli.

**Formal analysis:** Johannes Pilic, Benjamin Gottschalk, Yusuf C. Erdogan.

**Funding acquisition:** Wolfgang F. Graier, Roland Malli.

**Investigation:** Johannes Pilic, Furkan E. Oflaz, Benjamin Gottschalk.

**Methodology:** Johannes Pilic, Furkan E. Oflaz, Benjamin Gottschalk, Yusuf C. Erdogan.

**Project administration:** Wolfgang F. Graier.

**Supervision:** Wolfgang F. Graier, Roland Malli.

**Validation:** Johannes Pilic, Benjamin Gottschalk, Yusuf C. Erdogan.

**Visualization:** Johannes Pilic, Benjamin Gottschalk.

**Writing – original draft:** Johannes Pilic, Roland Malli.

**Writing – review & editing:** Johannes Pilic, Furkan E. Oflaz, Benjamin Gottschalk, Yusuf C. Erdogan, Wolfgang F. Graier, Roland Malli.

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
