## [Decision Letter · Decision Letter 0]

11 Mar 2024

PONE-D-24-04186Visualizing VDAC1 in live cells using a tetracysteine tagPLOS ONE

Dear Dr. Malli,

Thank you for submitting your manuscript to PLOS ONE. After careful consideration, we feel that it has merit but does not fully meet PLOS ONE’s publication criteria as it currently stands. Therefore, we invite you to submit a revised version of the manuscript that addresses the points raised during the review process. Please submit your revised manuscript by Apr 25 2024 11:59PM. If you will need more time than this to complete your revisions, please reply to this message or contact the journal office at plosone@plos.org. Please include the following items when submitting your revised manuscript:A rebuttal letter that responds to each point raised by the academic editor and reviewer(s). You should upload this letter as a separate file labeled 'Response to Reviewers'.A marked-up copy of your manuscript that highlights changes made to the original version. You should upload this as a separate file labeled 'Revised Manuscript with Track Changes'.An unmarked version of your revised paper without tracked changes. You should upload this as a separate file labeled 'Manuscript'.

We look forward to receiving your revised manuscript.

Kind regards,

Philipp J. Kahle, Ph.D.

Academic Editor

PLOS ONE

Journal Requirements:

   "The research was supported by the Molecular Medicine PhD program of the Medical University of Graz and the FWF (Austrian Science Fund: DK-MCD W1226 to W.F.G and I3716-B27 to R.M.)."  

Additional Editor Comments:

While this development of VDAC1 imaging tools is a welcome technical advance, both reviewers express the need of more careful validations, especially regarding the localization of VDAC1 to fission sites and contact points. Moreover, it is essential to document quantitative expression levels of the respective VDAC1 proteins, also in relation to endogenous.

Reviewers' comments:

Reviewer's Responses to Questions

**Comments to the Author**

1. Is the manuscript technically sound, and do the data support the conclusions?

Reviewer #1: Yes

Reviewer #2: Partly

2. Has the statistical analysis been performed appropriately and rigorously? 

Reviewer #1: Yes

Reviewer #2: No

3. Have the authors made all data underlying the findings in their manuscript fully available?

Reviewer #1: Yes

Reviewer #2: No

4. Is the manuscript presented in an intelligible fashion and written in standard English?

Reviewer #1: Yes

Reviewer #2: Yes

5. Review Comments to the Author

Reviewer #1: In this manuscript, the authors addressed the structural organization and localization in sub-cellular compartments of VDAC1 in living cells. To visualization VDAC1 dynamics in living cells, VDAC1 tagged by tetracysteine (TC) and its interaction with CYS-specific fluorogenic reagent were used, allowing its imaging. The results demonstrate that VDAC1 in the mitochondria has a cluster-like organization, and the majority of VDAC1-TC clusters were localized at the MAM. In addition, VDAC1-TC clusters were also identified in mitochondrial fission sites.

This work undoubtedly presents interesting and novel findings on VDAC1 clustering in certain cellular compartments, and points to VDAC1 function in ER–Mito structural and functional cross-talk, as well in mitochondria fission–fusion. However, the following points should be addressed before the manuscript can be accepted for publication.

Comments

1. It has been reported that both GFP-VDAC1 and VDAC1-GFP are predominantly aggregated in the cytosol. It should be noted, however, that VDAC1-GFP, but not GFP-VDAC1, was found to be targeted to the mitochondria and co-localized with Mito tracker, inducing cell death (VDAC1-GFP at the C-Terminus (see Aram, L., Geula, S., Arbel. N., and Shoshan-Barmatz, V. (2010) VDAC1 cysteine residues: Topology and function in channel activity and apoptosis. Biochemical J. 427, 445-454.).

Similarly, even the VDAC1 dimer fused to GFP (VDAC1-VDAC1-GFP) was found to be targeted to the mitochondria (Mader, A., Abu-Hamad, S., Arbel, N., Gutierrez-Aguilar, M., and Shoshan-Barmatz, V. (2010) Dominant-negative VDAC1 mutants reveal oligomeric VDAC1 to be active unit in mitochondria-mediated apoptosis, Biochemical J. 429(1):147-155, Fig. 2).

Aggregation of expressed VDAC1 was observed upon high expression levels. Thus, low plasmid concentration and/or short time transfection may prevent the high production and aggregation.

It is also possible that it is cell-type dependent, as this and other cited studies [3,4] showed poor targeting of C-terminal-tagged VDAC1 in HeLa cells, while the above studies used HEK-293T cells, as well as a different transfection agent and a tetracycline-inducible pcDNA4/TO vector.

I suggest indicating that, in other cells and using different transfection agents, VDAC1-GFP can be targeted to the mitochondria and that the mitochondria-targeting sequence is not in either the VDAC1 N- or C-terminus.

2. Using EM, enrichment of VDAC1 in the ER–Mito and Mito–Mito contact sites has been shown in the mouse cerebellum. This more physiological study supports the finding presented here, and this reviewer suggests indicating it (Shoshan-Barmatz, V., Zalk, R., Gincel, D., and Vardi, N. (2004) Subcellular localization of VDAC in cerebellum and its function in ER-Mitochondria cross-talk. Biochem. Biophys. Acta 1657, 105-114).

3. It will be very interesting to monitor changes in the distribution of VDAC1 clusters upon induction of apoptosis, ER stress, or inhibition of VDAC1 oligomerization. I believe the developed tool of TC-VDAC1 allows this, but it could be the topic of another study.

4. This reviewer could not find any results with Seahorse measurements. Although the method was presented, I may have missed it,

Minor comments:

1. HeLa S3—the meaning of S3 is not well known. Add what it indicates.

2. Line 95—add the composition of the storage buffer

3. Line 149—in the table, the plasmid in which GFP is fused to the C-terminus of VDAC1 is labelled as VDAC1-VDAC1 instead of VDAC1-GFP

4. Concluding sentence—"Since VDAC1-TC is the only tool to image this essential protein in living cells, we believe this approach will provide insights into the role of VDAC1 in cellular pathophysiology"—As indicated in comment 3 above, this tool can also be used under conditions leading to mitochondria dysfunction, apoptosis, autophagy, and more. This is more likely to be used for, rather than in, pathological conditions that are followed more in vivo.

Reviewer #2: Malli et al. in their manuscript entitled 'Visualizing VDAC1 in live cells using a tetracysteine tag' studied in HeLa cells the mitochondrial localization of the VDAC1 protein tagged with a tetracysteine sequence (TC-tag) expressed by transfection of a commercial construct. The work focuses on a useful tool to solve some known mislocalization problems of recombinant VDAC1 fused to fluorescent proteins. The important role played by VDAC1 in energy metabolism and cellular homeostasis justifies AA's interest in this protein.

The results obtained show the correct localization of VDAC1-TC at the outer mitochondrial membrane (OMM), mainly in the form of clusters and at the mitochondria-reticulum contact sites, confirming already known data. The use of a TC-tag to fluorescently follow the localization of a recombinant protein has also been known for some time. Furthermore, the study suffers from possible negative aspects of the methodology not evaluated by the authors. Also, the possible co-localization of VDAC1-TC to fission sites is not well supported.

Therefore, although the manuscript is clear, well-written and with high-resolution images, it is overall poor in new information of interest to the scientific community, not sufficient to justify its publication in PloS One.

Main comments:

1) Regarding the FlAsH-EDT2 and ReAsH-EDT2 dyes used to visualize TC-tag, there is no data in the manuscript on their half-life or efficacy over time. It should be verified whether the cells observed for hours or days continue to be visible. Given the available instrumentation, the authors should acquire real-time images at intervals of time and for a total time length of even days. This would also support the term 'dynamics' used somewhat improperly in the text.

2) Also, FlAsH-EDT2 and ReAsH-EDT2 are aromatic compounds, but they are not necessarily cationic, so their accumulation in the mitochondria may not be due solely to their aromatic nature. In fact, they bind non-specifically to endogenous cysteine-rich proteins. There is therefore a possibility that these dyes may target other cysteine-rich domains in various mitochondrial proteins. E.g. VDAC2 and VDAC3 are cysteine-rich proteins that are widely expressed, like the VDAC1 isoform, at the OMM. The authors should take this into consideration and provide appropriate control experiments.

3) Both fluorigenic dyes used are aromatic rings with delocalized electrons that, like TMRM, can generate reactive oxygen species (ROS) when irradiated, affecting Δψ. Therefore, the use of VDAC1-TC for monitoring mitochondrial interactions must be supported by experimental data. The authors should monitor the behaviour of VDAC1-TC over time using advanced fluorescence and perfusion microscopy tools at their disposal. They should also measure ROS following overexpression and labelling of VDAC1-TC.

4) The study would benefit greatly from a more thorough evaluation of the dye's impact on mitochondrial potential over time. TMRM is known to dynamically equilibrate inside and outside mitochondria based on fluctuations in mitochondrial membrane potential (Δψ) but Δψ is significantly influenced by the overall mitochondrial redox state. Furthermore, both FlAsH-EDT2 and ReAsH-EDT2 are similar not only to Rodh 123 but also to TMRM. Rodh 123 and TMRM accumulate in mitochondria according to the Nernst equation due to their positive net charge and -180 mV of Δψ.

5) The authors do not provide data to rule out a possible toxic effect related to VDAC1-TC overexpression. With a simple cell viability assay (like MTT), performed at different times, they could easily rule this out.

6) Data on the transfection efficiency of the VDAC1-TC-expressing construct are also lacking.

7) the TC-tag, due to its cysteine-rich sequence, could profoundly affect VDAC1 functionality. Electrophysiology experiments should be carried out to assess the effect of TC-tag on the functionality and voltage dependence of the VDAC1 channel.

8) Regarding the presence of VDAC at fission sites, colocalization with recognized actors in the process (such as Drp1) must be assessed. A proper investigation of this aspect would greatly benefit the study.

9) Statistical methods need to be reviewed. Technical replications alone are not sufficient when it comes to biological samples.

10) Which of the known and published structures of VDAC1 were modelled on the 3D structures of VDAC1 presented. It is known that at the N-terminus there is a mobile alpha helix, located at the cytosolic face of the pore or folded inside the channel, but in the presented figures this portion is not present. Furthermore, literature data show that the N- and C- terminal ends of VDAC1 are located on opposite sides of the channel (Tomasello MF et al. PLoS One (2013) 8, e81522), whereas in the manuscript figures they face the same side of the channel. Did the authors perhaps use VDAC1 constructs lacking the N-terminal alpha helix for their 3D analyses and/or in in vivo experiments?

11) Mitochondrial function is assessed by TMRM and Seahorse respirometry analysis (why not show flows?). In this respect, the authors should evaluate parameters other than Δψ and OCR/ECAR to avoid potential dye toxicity. Furthermore, since overexpression of VDAC1 itself is known to induce mitochondrial fragmentation, the use of this chimera in this context raises questions. Mitochondrial function and cell viability after VDAC1-TC overexpression and dye exposure must be evaluated over time.

12) The authors show that mitochondria are healthy after transfection with VDAC1-TC but then report hyperfragmentation of the mitochondria. This is strange.

Minor comments:

- Line 30: This statement should be revised because it is not entirely accurate. Many studies conducted in the last 20 years have utilized C- and N-terminal tags of VDAC isoforms, such as HA or Myc, which exceed 8 amino acids in size.

- in the table showing the plasmids used, VDAC1-VDAC1 should probably be replaced with VDAC1-GFP.

- throughout the text, the term 'dynamic' should be replaced with 'in live cell imaging'. There are in fact no data from dynamic localization analysis in the manuscript.

6. PLOS authors have the option to publish the peer review history of their article (what does this mean?). If published, this will include your full peer review and any attached files.

Reviewer #1: No

Reviewer #2: No

---

## [Author Response · Author response to Decision Letter 0]

8 Aug 2024

Dear Dr. Kahle,

We were delighted to learn that the reviewers found our study on visualizing VDAC1 in live cells interesting and that we have been given the opportunity to revise it. During this revision process, we conducted additional experiments and believe that we have successfully addressed all the main points raised by the three reviewers. Below is our Point-to-Point Response Letter, where we precisely describe and elucidate all the changes.

We also apologize for the extended duration it took us to complete this thorough revision. The acceptance of a new position at ETH Zurich by Johannes, the first author, resulted in some unavoidable delays. 

Nevertheless, we are now back on track and hopeful that the manuscript is now in a state where it can be accepted for publication. 

We greatly appreciate your continued support throughout this process.

Kind regards,

Roland Malli (on behalf of all authors)

Point-to-point rebuttal to the comments of the reviewers:

Reviewer #1 suggestions

In this manuscript, the authors addressed the structural organization and localization in sub-cellular compartments of VDAC1 in living cells. To visualization VDAC1 dynamics in living cells, VDAC1 tagged by tetracysteine (TC) and its interaction with CYS-specific fluorogenic reagent were used, allowing its imaging. The results demonstrate that VDAC1 in the mitochondria has a cluster-like organization, and the majority of VDAC1-TC clusters were localized at the MAM. In addition, VDAC1-TC clusters were also identified in mitochondrial fission sites.

This work undoubtedly presents interesting and novel findings on VDAC1 clustering in certain cellular compartments, and points to VDAC1 function in ER–Mito structural and functional cross-talk, as well in mitochondria fission–fusion. However, the following points should be addressed before the manuscript can be accepted for publication.

Response: We are delighted that the reviewer found our findings interesting and novel. We have carefully considered the feedback and have made the suggested revisions to improve the quality of our paper.

Comments

1. It has been reported that both GFP-VDAC1 and VDAC1-GFP are predominantly aggregated in the cytosol. It should be noted, however, that VDAC1-GFP, but not GFP-VDAC1, was found to be targeted to the mitochondria and co-localized with Mito tracker, inducing cell death (VDAC1-GFP at the C-Terminus (see Aram, L., Geula, S., Arbel. N., and Shoshan-Barmatz, V. (2010) VDAC1 cysteine residues: Topology and function in channel activity and apoptosis. Biochemical J. 427, 445-454.).

Similarly, even the VDAC1 dimer fused to GFP (VDAC1-VDAC1-GFP) was found to be targeted to the mitochondria (Mader, A., Abu-Hamad, S., Arbel, N., Gutierrez-Aguilar, M., and Shoshan-Barmatz, V. (2010) Dominant-negative VDAC1 mutants reveal oligomeric VDAC1 to be active unit in mitochondria-mediated apoptosis, Biochemical J. 429(1):147-155, Fig. 2).

Aggregation of expressed VDAC1 was observed upon high expression levels. Thus, low plasmid concentration and/or short time transfection may prevent the high production and aggregation.

It is also possible that it is cell-type dependent, as this and other cited studies [3,4] showed poor targeting of C-terminal-tagged VDAC1 in HeLa cells, while the above studies used HEK-293T cells, as well as a different transfection agent and a tetracycline-inducible pcDNA4/TO vector.

I suggest indicating that, in other cells and using different transfection agents, VDAC1-GFP can be targeted to the mitochondria and that the mitochondria-targeting sequence is not in either the VDAC1 N- or C-terminus.

Response: We thank the reviewer for these suggestions. As suggested by the reviewer, we have added the references and discussed that the observed mitochondrial localization of rat VDAC1-GFP could be due to different cell-type, transfection agent, or vector used in these studies (p.13, lines 285-288). As suggested by the reviewer, we have revised our experimental approach to include lower plasmid concentrations to mitigate the high production and aggregation of VDAC1. We reduced the amount of plasmid DNA used for transfection from 0.5 µg to 0.166 µg. Using less plasmid DNA to reduce potential overexpression artifacts, we observed a reduced percentage of cells with aggregated FP-tagged VDAC1 (Fig 1B), but we did not observe mitochondrial localization (Fig S1B). This suggest, that in HeLa cells, GFP-tagged VDAC1 aggregates in the cytosol and does not localize to mitochondria.

2. Using EM, enrichment of VDAC1 in the ER–Mito and Mito–Mito contact sites has been shown in the mouse cerebellum. This more physiological study supports the finding presented here, and this reviewer suggests indicating it (Shoshan-Barmatz, V., Zalk, R., Gincel, D., and Vardi, N. (2004) Subcellular localization of VDAC in cerebellum and its function in ER-Mitochondria cross-talk. Biochem. Biophys. Acta 1657, 105-114).

Response: We thank the reviewer for pointing us to this more physiological study. We have added this study to our references (ref. 15).

3. It will be very interesting to monitor changes in the distribution of VDAC1 clusters upon induction of apoptosis, ER stress, or inhibition of VDAC1 oligomerization. I believe the developed tool of TC-VDAC1 allows this, but it could be the topic of another study.

Response: We thank the reviewer for these suggestion. We monitored changes in the distribution of VDAC1 clusters upon induction of apoptosis, as suggested by the reviewer. We found that approxematly 25 - 40% VDAC1-TC clusters colocalize with BAK-clusters, a major player in mitochondrial apoptosis, during induction of apoptosis (new Fig 4 and Fig S4). While we have focused on the current experiments on the induction of apoptosis, we acknowledge that the induction of ER stress or the inhibitor of VDAC1 oligomerization could be an interesting topic for future studies using the TC-VDAC1 tool.

4. This reviewer could not find any results with Seahorse measurements. Although the method was presented, I may have missed it.

Response: The analysis of the Seahorse measurements can be found in the supplementary figures (Fig S2D).

Minor comments:

1. HeLa S3—the meaning of S3 is not well known. Add what it indicates.

Response: We have added the meaning of HeLa S3 (p3., line 45).

2. Line 95—add the composition of the storage buffer

Response: The composition of storage buffer can be found on p3., lines 59-63.

3. Line 149—in the table, the plasmid in which GFP is fused to the C-terminus of VDAC1 is labelled as VDAC1-VDAC1 instead of VDAC1-GFP

Response: We thank the reviewer for pointing out our mistake. We have corrected it.

4. Concluding sentence—"Since VDAC1-TC is the only tool to image this essential protein in living cells, we believe this approach will provide insights into the role of VDAC1 in cellular pathophysiology"—As indicated in comment 3 above, this tool can also be used under conditions leading to mitochondria dysfunction, apoptosis, autophagy, and more. This is more likely to be used for, rather than in, pathological conditions that are followed more in vivo.

Response: We agree with the reviewer. We have replaced the term “pathophysiology” with “processes” to reflect a more likely use of this tool (p.14, line 315).

Reviewer #2 suggestions

Malli et al. in their manuscript entitled 'Visualizing VDAC1 in live cells using a tetracysteine tag' studied in HeLa cells the mitochondrial localization of the VDAC1 protein tagged with a tetracysteine sequence (TC-tag) expressed by transfection of a commercial construct. The work focuses on a useful tool to solve some known mislocalization problems of recombinant VDAC1 fused to fluorescent proteins. The important role played by VDAC1 in energy metabolism and cellular homeostasis justifies AA's interest in this protein.

The results obtained show the correct localization of VDAC1-TC at the outer mitochondrial membrane (OMM), mainly in the form of clusters and at the mitochondria-reticulum contact sites, confirming already known data. The use of a TC-tag to fluorescently follow the localization of a recombinant protein has also been known for some time. Furthermore, the study suffers from possible negative aspects of the methodology not evaluated by the authors. Also, the possible co-localization of VDAC1-TC to fission sites is not well supported.

Therefore, although the manuscript is clear, well-written and with high-resolution images, it is overall poor in new information of interest to the scientific community, not sufficient to justify its publication in PloS One.

Response: We thank the reviewer for appreciating the clarity and quality of our manuscript clear. We agree with the reviewer that while the TC tag has been known for some time, deciphering how VDAC1 needs to be tagged to be both visualizable and functional in live cells was not known. We respectfully disagree with the assessment that the tool is of limited interest, as it already has been used in collaborative research and is frequently requested on addgene.com, indicating its relevance to the scientific community. As suggested by the reviewer, we have incorporated additional methods to enhance the quality of our manuscript.

Main comments:

1) Regarding the FlAsH-EDT2 and ReAsH-EDT2 dyes used to visualize TC-tag, there is no data in the manuscript on their half-life or efficacy over time. It should be verified whether the cells observed for hours or days continue to be visible. Given the available instrumentation, the authors should acquire real-time images at intervals of time and for a total time length of even days. This would also support the term 'dynamics' used somewhat improperly in the text.

Response: We thank the reviewer for the suggestion. We agree about the importance in demonstrating the efficacy of our tool over time. As suggested by the reviewer, we visualized ReAsH-labeled VDAC1-TC with high temporal resolution (Video 1) and FlAsH-labeled VDAC1-TC over hours (Video 2), indicating that our tool is suitable for dynamic measurements. We have added a description of these results (p.10, lines 212-216).

2) Also, FlAsH-EDT2 and ReAsH-EDT2 are aromatic compounds, but they are not necessarily cationic, so their accumulation in the mitochondria may not be due solely to their aromatic nature. In fact, they bind non-specifically to endogenous cysteine-rich proteins. There is therefore a possibility that these dyes may target other cysteine-rich domains in various mitochondrial proteins. E.g. VDAC2 and VDAC3 are cysteine-rich proteins that are widely expressed, like the VDAC1 isoform, at the OMM. The authors should take this into consideration and provide appropriate control experiments.

Response: We thank the reviewer for pointing out this possibility. We have described the possibility of the dyes to non-specifically bind cysteine-rich proteins (p. 9, lines 189-191). Without VDAC1-TC overexpression we did not observe a cluster-like distribution of FlAsH (Fig S2A), suggesting that the dye alone does not cluster on mitochondria. 

3) Both fluorigenic dyes used are aromatic rings with delocalized electrons that, like TMRM, can generate reactive oxygen species (ROS) when irradiated, affecting Δψ. Therefore, the use of VDAC1-TC for monitoring mitochondrial interactions must be supported by experimental data. The authors should monitor the behaviour of VDAC1-TC over time using advanced fluorescence and perfusion microscopy tools at their disposal. They should also measure ROS following overexpression and labelling of VDAC1-TC.

Response: We thank the reviewer for these insights. As suggested by the reviewer, we have measured mitochondrial ROS production when FlAsH stained VDAC1-TC overexpressing cells were irridiated over time. Our measurements show that FlAsH staining in VDAC1-TC overexpressing cells did not lead to ROS generation during irradiation over time (Fig S2G).

4) The study would benefit greatly from a more thorough evaluation of the dye's impact on mitochondrial potential over time. TMRM is known to dynamically equilibrate inside and outside mitochondria based on fluctuations in mitochondrial membrane potential (Δψ) but Δψ is significantly influenced by the overall mitochondrial redox state. Furthermore, both FlAsH-EDT2 and ReAsH-EDT2 are similar not only to Rodh 123 but also to TMRM. Rodh 123 and TMRM accumulate in mitochondria according to the Nernst equation due to their positive net charge and -180 mV of Δψ.

Response: We agree with the reviewer that our study would benefit from a more thorough evaluation of the dye's impact on mitochondrial potential over time. As suggested by the reviewer, we have measured mitochondrial membrane potential when FlAsH stained cells were irridiated over time. Our measurements show that FlAsH staining did not affect mitochondrial membrane potential over time, compared to unstained cells (Fig S2E), indicating that FlAsH staining does not affect ROS production and thus mitochondrial membrane potential during time-lapse imaging.

5) The authors do not provide data to rule out a possible toxic effect related to VDAC1-TC overexpression. With a simple cell viability assay (like MTT), performed at different times, they could easily rule this out.

Response: We thank the reviewer for the suggestion. We assessed whether VDAC1-TC overexpression affects cell proliferation, as a marker of cellular health, over time. We observed that cells overexpressing VDAC1-TC exhibited reduced cellular proliferation compared to sham-transfected cells (Fig S5B). These data indicate that the hyperfragmentation induced by VDAC1-TC overexpression reduces cellular fitness.

6) Data on the transfection efficiency of the VDAC1-TC-expressing construct are also lacking.

Response: We thank the reviewer for the comment. We have included data on the transfection efficiency of the VDAC1-TC construct, which is approximately 50% (Fig S5C).

7) the TC-tag, due to its cysteine-rich sequence, could profoundly affect VDAC1 functionality. Electrophysiology experiments should be carried out to assess the effect of TC-tag on the functionality and voltage dependence of the VDAC1 channel.

Response: We thank the reviewer for the comment. We agree that assessing the impact of the TC-tag on VDAC1 functionality is important. However, conducting electrophysiology experiments to evaluate the effect of the TC-tag on VDAC1's functionality and voltage dependence is beyond the scope and current feasibility of our study. Nonetheless, we have discussed the potential impact of the TC-tag on VDAC1 functionality in the revised manuscript on p.14, lines 312-313.

8) Regarding the presence of VDAC at fission sites, colocalization with recognized actors in the process (such as Drp1) must be assessed. A proper investigation of this aspect would greatly benefit the study.

Response: We thank the reviewer for this valuable suggestion. We agree that assessing colocalization of VDAC with known fission-related proteins would provide additional insights. As suggested by the reviewer, we assessed the presence of VDAC1-TC clusters at fission sites with the mitochondrial fission receptor mitochondrial fission factor (Mff), using mCh-Mff as a marker. We observed that VDAC1-TC-clusters were localized to Mff-clusters when cells were treated with FCCP (Fig S5D). These data indicate that VDAC1 might be involved in mitochondrial fission.

9) Statistical methods need to be reviewed. Technical replications alone are not sufficient when it comes to biological samples.

Response: We thank the reviewer for the feedback. We have conducted multiple independent experiments on different days to account for biological variability, which we believe provides a robust basis for our conclusions. We feel that the replicates included are sufficient to support the statistical analysis presented in our study.

10) Which of the known and published structures of VDAC1 were modelled on the 3D structures of VDAC1 presented. It is known that at the N-terminus there is a mobile alpha helix, located at the cytosolic face of the pore or folded inside the channel, but in the presented figures this portion is not present. Furthermore, literature data show that the N- and C- terminal e

---

## [Decision Letter · Decision Letter 1]

13 Sep 2024

Visualizing VDAC1 in live cells using a tetracysteine tag

PONE-D-24-04186R1

Dear Dr. Malli,

We’re pleased to inform you that your manuscript has been judged scientifically suitable for publication and will be formally accepted for publication once it meets all outstanding technical requirements.

Kind regards,

Philipp J. Kahle, Ph.D.

Academic Editor

PLOS ONE

Additional Editor Comments (optional):

Reviewers' comments:

Reviewer's Responses to Questions

**Comments to the Author**

1. If the authors have adequately addressed your comments raised in a previous round of review and you feel that this manuscript is now acceptable for publication, you may indicate that here to bypass the “Comments to the Author” section, enter your conflict of interest statement in the “Confidential to Editor” section, and submit your "Accept" recommendation.

Reviewer #1: All comments have been addressed

2. Is the manuscript technically sound, and do the data support the conclusions?

Reviewer #1: Yes

3. Has the statistical analysis been performed appropriately and rigorously? 

Reviewer #1: Yes

4. Have the authors made all data underlying the findings in their manuscript fully available?

Reviewer #1: Yes

5. Is the manuscript presented in an intelligible fashion and written in standard English?

Reviewer #1: Yes

6. Review Comments to the Author

Reviewer #1: No comments,

The authors have addressed all my concerns in a satisfactory manner. Thus, the paper is now can be accepted for publications.

7. PLOS authors have the option to publish the peer review history of their article (what does this mean?). If published, this will include your full peer review and any attached files.

Reviewer #1: **Yes: **Varda Shoshan-Barmatz

---

## [Editor Report · Acceptance letter]

8 Oct 2024

PONE-D-24-04186R1 

PLOS ONE

Dear Dr. Malli, 

I'm pleased to inform you that your manuscript has been deemed suitable for publication in PLOS ONE. Congratulations! Your manuscript is now being handed over to our production team.

Kind regards, 

on behalf of

Prof. Philipp J. Kahle 

Academic Editor

PLOS ONE